# Clustering with Same-Cluster Queries

**Hassan Ashtiani** , **Shrinu Kushagra** and **Shai Ben-David**
David R. Cheriton School of Computer Science
University of Waterloo,
Waterloo, Ontario, Canada
{mhzokaei,skushagr,shai}@uwaterloo.ca

## Abstract

We propose a framework for Semi-Supervised Active Clustering framework
(SSAC), where the learner is allowed to interact with a domain expert, asking
whether two given instances belong to the same cluster or not. We study the query
and computational complexity of clustering in this framework. We consider a
setting where the expert conforms to a center-based clustering with a notion of
margin. We show that there is a trade off between computational complexity and
query complexity; We prove that for the case of $k$-means clustering (i.e., when the
expert conforms to a solution of $k$-means), having access to relatively few such
queries allows efficient solutions to otherwise NP hard problems.

In particular, we provide a probabilistic polynomial-time (BPP) algorithm for
clustering in this setting that asks $O\big(k^2 \log k + k \log n\big)$ same-cluster queries and
runs with time complexity $O\big(kn \log n\big)$ (where $k$ is the number of clusters and
$n$ is the number of instances). The algorithm succeeds with high probability for
data satisfying margin conditions under which, without queries, we show that the
problem is NP hard. We also prove a lower bound on the number of queries needed
to have a computationally efficient clustering algorithm in this setting.

## 1 Introduction

Clustering is a challenging task particularly due to two impediments. The first problem is that
clustering, in the absence of domain knowledge, is usually an *under-specified* task; the solution
of choice may vary significantly between different intended applications. The second one is that
performing clustering under many natural models is computationally hard.

Consider the task of dividing the users of an online shopping service into different groups. The result
of this clustering can then be used for example in suggesting similar products to the users in the same
group, or for organizing data so that it would be easier to read/analyze the monthly purchase reports.
Those different applications may result in conflicting solution requirements. In such cases, one needs
to exploit domain knowledge to better define the clustering problem.

Aside from trial and error, a principled way of extracting domain knowledge is to perform clustering
using a form of 'weak' supervision. For example, Balcan and Blum [BB08] propose to use an
interactive framework with 'split/merge' queries for clustering. In another work, Ashtiani and
Ben-David [ABD15] require the domain expert to provide the clustering of a 'small' subset of data.

At the same time, mitigating the computational problem of clustering is critical. Solving most of
the common optimization formulations of clustering is NP-hard (in particular, solving the popular
$k$-means and $k$-median clustering problems). One approach to address this issues is to exploit the
fact that natural data sets usually exhibit some nice properties and likely to avoid the worst-case
scenarios. In such cases, optimal solution to clustering may be found efficiently. The quest for notions

of niceness that are likely to occur in real data and allow clustering efficiency is still ongoing (see [Ben15] for a critical survey of work in that direction).

In this work, we take a new approach to alleviate the computational problem of clustering. In particular, we ask the following question: can weak supervision (in the form of answers to natural queries) help relaxing the computational burden of clustering? This will add up to the other benefit of supervision: making the clustering problem better defined by enabling the accession of domain knowledge through the supervised feedback.

The general setting considered in this work is the following. Let $X$ be a set of elements that should be clustered and $d$ a dissimilarity function over it. The oracle (e.g., a domain expert) has some information about a target clustering $C_X^*$ in mind. The clustering algorithm has access to $X, d$, and can also make queries about $C_X^*$. The queries are in the form of *same-cluster* queries. Namely, the algorithm can ask whether two elements belong to the same cluster or not. The goal of the algorithm is to find a clustering that meets some predefined clusterability conditions and is consistent with the answers given to its queries.

We will also consider the case that the oracle conforms with some optimal $k$-means solution. We then show that access to a 'reasonable' number of same-cluster queries can enable us to provide an efficient algorithm for otherwise NP-hard problems.

## 1.1  Contributions

The two main contributions of this paper are the introduction of the semi-supervised active clustering (SSAC) framework and, the rather unusual demonstration that access to simple query answers can turn an otherwise NP hard clustering problem into a feasible one.

Before we explain those results, let us also mention a notion of clusterability (or 'input niceness') that we introduce. We define a novel notion of niceness of data, called $\gamma$-margin property that is related to the previously introduced notion of center proximity [ABS12]. The larger the value of $\gamma$, the stronger the assumption becomes, which means that clustering becomes easier. With respect to that $\gamma$ parameter, we get a sharp 'phase transition' between $k$-means being NP hard and being optimally solvable in polynomial time[1].

We focus on the effect of using queries on the computational complexity of clustering. We provide a probabilistic polynomial time (BPP) algorithm for clustering with queries, that succeeds under the assumption that the input satisfies the $\gamma$-margin condition for $\gamma > 1$. This algorithm makes $O(k^2 \log k + k \log n)$ same-cluster queries to the oracle and runs in $O(kn \log n)$ time, where $k$ is the number of clusters and $n$ is the size of the instance set.

On the other hand, we show that without access to query answers, $k$-means clustering is NP-hard even when the solution satisfies $\gamma$-margin property for $\gamma = \sqrt{3.4} \approx 1.84$ and $k = \Theta(n^\epsilon)$ (for any $\epsilon \in (0, 1)$). We further show that access to $\Omega(\log k + \log n)$ queries is needed to overcome the NP hardness in that case. These results, put together, show an interesting phenomenon. Assume that the oracle conforms to an optimal solution of $k$-means clustering and that it satisfies the $\gamma$-margin property for some $1 < \gamma \leq \sqrt{3.4}$. In this case, our lower bound means that without making queries $k$-means clustering is NP-hard, while the positive result shows that with a reasonable number of queries the problem becomes efficiently solvable.

This indicates an interesting (and as far as we are aware, novel) trade-off between query complexity and computational complexity in the clustering domain.

## 1.2  Related Work

This work combines two themes in clustering research; clustering with partial supervision (in particular, supervision in the form of answers to queries) and the computational complexity of clustering tasks.

Supervision in clustering (sometimes also referred to as 'semi-supervised clustering') has been addressed before, mostly in application-oriented works [BBM02, BBM04, KBDM09]. The most

common method to convey such supervision is through a set of pairwise *link/do-not-link* constraints on the instances. Note that in contrast to the supervision we address here, in the setting of the papers cited above, the supervision is non-interactive. On the theory side, Balcan et. al [BB08] propose a framework for interactive clustering with the help of a user (i.e., an oracle). The queries considered in that framework are different from ours. In particular, the oracle is provided with the current clustering, and tells the algorithm to either split a cluster or merge two clusters. Note that in that setting, the oracle should be able to evaluate the whole given clustering for each query.

Another example of the use of supervision in clustering was provided by Ashtiani and Ben-David [ABD15]. They assumed that the target clustering can be approximated by first mapping the data points into a new space and then performing $k$-means clustering. The supervision is in the form of a clustering of a small subset of data (the subset provided by the learning algorithm) and is used to search for such a mapping.

Our proposed setup combines the user-friendliness of *link/don't-link* queries (as opposed to asking the domain expert to answer queries about whole data set clustering, or to cluster sets of data) with the advantages of interactiveness.

The computational complexity of clustering has been extensively studied. Many of these results are negative, showing that clustering is computationally hard. For example, $k$-means clustering is NP-hard even for $k = 2$ [Das08], or in a 2-dimensional plane [Vat09, MNV09]. In order to tackle the problem of computational complexity, some notions of niceness of data under which the clustering becomes easy have been considered (see [Ben15] for a survey).

The closest proposal to this work is the notion of $\alpha$-center proximity introduced by Awasthi et. al [ABS12]. We discuss the relationship of that notion to our notion of margin in Appendix B. In the restricted scenario (i.e., when the centers of clusters are selected from the data set), their algorithm efficiently recovers the target clustering (outputs a tree such that the target is a pruning of the tree) for $\alpha > 3$. Balcan and Liang [BL12] improve the assumption to $\alpha > \sqrt{2} + 1$. Ben-David and Reyzin [BDR14] show that this problem is NP-Hard for $\alpha < 2$.

Variants of these proofs for our $\gamma$-margin condition yield the feasibility of $k$-means clustering when the input satisfies the condition with $\gamma > 2$ and NP hardness when $\gamma < 2$, both in the case of arbitrary (not necessarily Euclidean) metrics[2] .

## 2 Problem Formulation

### 2.1 Center-based clustering

The framework of clustering with queries can be applied to any type of clustering. However, in this work, we focus on a certain family of common clusterings – center-based clustering in Euclidean spaces[3].

Let $\mathcal{X}$ be a subset of some Euclidean space, $\mathbb{R}^d$. Let $\mathcal{C}_\mathcal{X} = \{C_1, \ldots, C_k\}$ be a clustering (i.e., a partitioning) of $\mathcal{X}$. We say $x_1 \overset{C_\mathcal{X}}{\sim} x_2$ if $x_1$ and $x_2$ belong to the same cluster according to $C_\mathcal{X}$. We further denote by $n$ the number of instances ($|\mathcal{X}|$) and by $k$ the number of clusters.

We say that a clustering $C_\mathcal{X}$ is *center-based* if there exists a set of centers $\mu = \{\mu_1, \ldots, \mu_k\} \subset \mathcal{R}^n$ such that the clustering corresponds to the Voroni diagram over those center points. Namely, for every $x$ in $\mathcal{X}$ and $i \leq k$, $x \in C_i \Leftrightarrow i = \arg\min_j d(x, \mu_j)$.

Finally, we assume that the centers $\mu^*$ corresponding to $C^*$ are the centers of mass of the corresponding clusters. In other words, $\mu_i^* = \frac{1}{|C_i|} \sum_{x \in C_i^*} x$. Note that this is the case for example when the oracle's clustering is the optimal solution to the Euclidean k-means clustering problem.

### 2.2 The $\gamma$-margin property

Next, we introduce a notion of clusterability of a data set, also referred to as 'data niceness property'.

**Definition 1** ($\gamma$-margin). *Let $\mathcal{X}$ be set of points in metric space $M$. Let $\mathcal{C}_\mathcal{X} = \{C_1, \ldots, C_k\}$ be a center-based clustering of $\mathcal{X}$ induced by centers $\mu_1, \ldots, \mu_k \in M$. We say that $\mathcal{C}_\mathcal{X}$ satisfies the $\gamma$-margin property if the following holds. For all $i \in [k]$ and every $x \in C_i$ and $y \in \mathcal{X} \setminus C_i$,*

$$\gamma d(x, \mu_i) < d(y, \mu_i)$$

Similar notions have been considered before in the clustering literature. The closest one to our $\gamma$-margin is the notion of $\alpha$-center proximity [BL12, ABS12]. We discuss the relationship between these two notions in appendix B.

## 2.3  The algorithmic setup

For a clustering $C^* = \{C_1^*, \ldots C_k^*\}$, a $C^*$-oracle is a function $\mathcal{O}_{C^*}$ that answers queries according to that clustering. One can think of such an oracle as a user that has some idea about its desired clustering, enough to answer the algorithm's queries. The clustering algorithm then tries to recover $C^*$ by querying a $C^*$-oracle. The following notion of query is arguably most intuitive.

**Definition 2** (Same-cluster Query). *A same-cluster query asks whether two instances $x_1$ and $x_2$ belong to the same cluster, i.e.,*

$$\mathcal{O}_{C^*}(x_1, x_2) = \begin{cases} true & if \ x_1 \overset{C^*}{\sim} x_2 \\ false & o.w. \end{cases}$$

*(we omit the subscript $C^*$ when it is clear from the context).*

**Definition 3** (Query Complexity). *An SSAC instance is determined by the tuple $(\mathcal{X}, d, C^*)$. We will consider families of such instances determined by niceness conditions on their oracle clusterings $C^*$.*

1. *A SSAC algorithm $\mathcal{A}$ is called a $q$-solver for a family $G$ of such instances, if for every instance $(\mathcal{X}, d, C^*) \in G$, it can recover $C^*$ by having access to $(\mathcal{X}, d)$ and making at most $q$ queries to a $C^*$-oracle.*

2. *Such an algorithm is a polynomial $q$-solver if its time-complexity is polynomial in $|\mathcal{X}|$ and $|C^*|$ (the number of clusters).*

3. *We say $G$ admits an $O(q)$ query complexity if there exists an algorithm $\mathcal{A}$ that is a polynomial $q$-solver for every clustering instance in $G$.*

# 3  An Efficient SSAC Algorithm

In this section we provide an efficient algorithm for clustering with queries. The setting is the one described in the previous section. In particular, it is assumed that the oracle has a center-based clustering in his mind which satisfies the $\gamma$-margin property. The space is Euclidean and the center of each cluster is the center of mass of the instances in that cluster. The algorithm not only makes same-cluster queries, but also another type of query defined as below.

**Definition 4** (Cluster-assignment Query). *A cluster-assignment query asks the cluster index that an instance $x$ belongs to. In other words $\mathcal{O}_{C^*}(x) = i$ if and only if $x \in C_i^*$.*

Note however that each cluster-assignment query can be replaced with $k$ same-cluster queries (see appendix A in supplementary material). Therefore, we can express everything in terms of the more natural notion of same-cluster queries, and the use of cluster-assignment query is just to make the representation of the algorithm simpler.

Intuitively, our proposed algorithm does the following. In the first phase, it tries to approximate the center of one of the clusters. It does this by asking cluster-assignment queries about a set of randomly (uniformly) selected point, until it has a sufficient number of points from at least one cluster (say $C_p$). It uses the mean of these points, $\mu_p'$, to approximate the cluster center.

In the second phase, the algorithm recovers all of the instances belonging to $C_p$. In order to do that, it first sorts all of the instances based on their distance to $\mu_p'$. By showing that all of the points in $C_p$ lie inside a sphere centered at $\mu_p'$ (which does not include points from any other cluster), it tries to find

the radius of this sphere by doing binary search using same-cluster queries. After that, the elements in $C_p$ will be located and can be removed from the data set. The algorithm repeats this process $k$ times to recover all of the clusters.

The details of our approach is stated precisely in Algorithm 1. Note that $\beta$ is a small constant[4]. Theorem 7 shows that if $\gamma > 1$ then our algorithm recovers the target clustering with high probability. Next, we give bounds on the time and query complexity of our algorithm. Theorem 8 shows that our approach needs $O(k \log n + k^2 \log k)$ queries and runs with time complexity $O(kn \log n)$.

---

**Algorithm 1:** Algorithm for $\gamma(> 1)$-margin instances with queries

---

**Input**: Clustering instance $\mathcal{X}$, oracle $\mathcal{O}$, the number of clusters $k$ and parameter $\delta \in (0, 1)$
**Output**: A clustering $\mathcal{C}$ of the set $\mathcal{X}$

$\mathcal{C} = \{\}, \mathcal{S}_1 = \mathcal{X}, \eta = \beta \frac{\log k + \log(1/\delta)}{(\gamma - 1)^4}$
**for** $i = 1$ *to* $k$ **do**

    **Phase 1**
    $l = k\eta + 1$;
    $Z \sim U^l[\mathcal{S}_i]$   // Draws $l$ independent elements from $\mathcal{S}_i$ uniformly at random
    For $1 \leq t \leq i$,
      $Z_t = \{x \in Z : \mathcal{O}(x) = t\}$.   //Asks cluster-assignment queries about the members of $Z$
    $p = \arg\max_t |Z_t|$
    $\mu'_p := \frac{1}{|Z_p|} \sum_{x \in Z_p} x$.

    **Phase 2**
    // We know that there exists $r_i$ such that $\forall x \in \mathcal{S}_i, x \in C_i \Leftrightarrow d(x, \mu'_i) < r_i$.
    // Therefore, $r_i$ can be found by simple binary search
    $\widehat{\mathcal{S}}_i = \text{Sorted}(\{\mathcal{S}_i\})$   // Sorts elements of $\{x : x \in \mathcal{S}_i\}$ in increasing order of $d(x, \mu'_p)$.
    $r_i = \text{BinarySearch}(\widehat{\mathcal{S}}_i)$   //This step takes up to $O(\log |\mathcal{S}_i|)$ same-cluster queries
    $C'_p = \{x \in \mathcal{S}_i : d(x, \mu'_p) \leq r_i\}$.
    $S_{i+1} = S_i \setminus C'_p$.
    $\mathcal{C} = \mathcal{C} \cup \{C'_p\}$
**end**

---

**Lemma 5.** *Let $(\mathcal{X}, d, C)$ be a clustering instance, where $C$ is center-based and satisfies the $\gamma$-margin property. Let $\mu$ be the set of centers corresponding to the centers of mass of $C$. Let $\mu'_i$ be such that $d(\mu_i, \mu'_i) \leq r(C_i)\epsilon$, where $r(C_i) = \max_{x \in C_i} d(x, \mu_i)$. Then $\gamma \geq 1 + 2\epsilon$ implies that*

$$\forall x \in C_i, \forall y \in \mathcal{X} \setminus C_i \Rightarrow d(x, \mu'_i) < d(y, \mu'_i)$$

*Proof.* Fix any $x \in C_i$ and $y \in C_j$. $d(x, \mu'_i) \leq d(x, \mu_i) + d(\mu_i, \mu'_i) \leq r(C_i)(1 + \epsilon)$. Similarly, $d(y, \mu'_i) \geq d(y, \mu_i) - d(\mu_i, \mu'_i) > (\gamma - \epsilon)r(C_i)$. Combining the two, we get that $d(x, \mu'_i) < \frac{1+\epsilon}{\gamma - \epsilon} d(y, \mu'_i)$. □

**Lemma 6.** *Let the framework be as in Lemma 5. Let $Z_p, C_p, \mu_p, \mu'_p$ and $\eta$ be defined as in Algorhtm 1, and $\epsilon = \frac{\gamma - 1}{2}$. If $|Z_p| > \eta$, then the probability that $d(\mu_p, \mu'_p) > r(C_p)\epsilon$ is at most $\frac{\delta}{k}$.*

*Proof.* Define a uniform distribution $U$ over $C_p$. Then $\mu_p$ and $\mu'_p$ are the true and empirical mean of this distribution. Using a standard concentration inequality (Thm. 12 from Appendix D) shows that the empirical mean is close to the true mean, completing the proof.

□

**Theorem 7.** *Let $(\mathcal{X}, d, C)$ be a clustering instance, where $C$ is center-based and satisfies the $\gamma$-margin property. Let $\mu_i$ be the center of mass of $C_i$. Assume $\delta \in (0, 1)$ and $\gamma > 1$. Then with probability at least $1 - \delta$, Algorithm 1 outputs $C$.*

*Proof.* In the first phase of the algorithm we are making $l > k\eta$ cluster-assignment queries. Therefore, using the pigeonhole principle, we know that there exists cluster index $p$ such that $|Z_p| > \eta$. Then Lemma 6 implies that the algorithm chooses a center $\mu'_p$ such that with probability at least $1 - \frac{\delta}{k}$ we have $d(\mu_p, \mu'_p) \le r(C_p)\epsilon$. By Lemma 5, this would mean that $d(x, \mu'_p) < d(y, \mu'_p)$ for all $x \in C_p$ and $y \notin C_p$. Hence, the radius $r_i$ found in the phase two of Alg. 1 is such that $r_i = \max_{x \in C_p} d(x, \mu'_p)$.

This implies that $C'_p$ (found in phase two) equals to $C_p$. Hence, with probability at least $1 - \frac{\delta}{k}$ one iteration of the algorithm successfully finds all the points in a cluster $C_p$. Using union bound, we get that with probability at least $1 - k\frac{\delta}{k} = 1 - \delta$, the algorithm recovers the target clustering. □

**Theorem 8.** *Let the framework be as in theorem 7. Then Algorithm 1*

- *Makes $O\big(k \log n + k^2 \frac{\log k + \log(1/\delta)}{(\gamma-1)^4}\big)$ same-cluster queries to the oracle $\mathcal{O}$.*
- *Runs in $O\big(kn \log n + k^2 \frac{\log k + \log(1/\delta)}{(\gamma-1)^4}\big)$ time.*

*Proof.* In each iteration (i) the first phase of the algorithm takes $O(\eta)$ time and makes $\eta + 1$ cluster-assignment queries (ii) the second phase takes $O(n \log n)$ times and makes $O(\log n)$ same-cluster queries. Each cluster-assignment query can be replaced with $k$ same-cluster queries; therefore, each iteration runs in $O(k\eta + n \log n)$ and uses $O(k\eta + \log n)$ same-cluster queries. By replacing $\eta = \beta \frac{\log k + \log(1/\delta)}{(\gamma-1)^4}$ and noting that there are $k$ iterations, the proof will be complete. □

**Corollary 9.** *The set of Euclidean clustering instances that satisfy the $\gamma$-margin property for some $\gamma > 1$ admits query complexity $O\big(k \log n + k^2 \frac{\log k + \log(1/\delta)}{(\gamma-1)^4}\big)$.*

## 4 Hardness Results

### 4.1 Hardness of Euclidean $k$-means with Margin

Finding $k$-means solution without the help of an oracle is generally computationally hard. In this section, we will show that solving Euclidean $k$-means remains hard even if we know that the optimal solution satisfies the $\gamma$-margin property for $\gamma = \sqrt{3.4}$. In particular, we show the hardness for the case of $k = \Theta(n^\epsilon)$ for any $\epsilon \in (0, 1)$.

In Section 3, we proposed a polynomial-time algorithm that could recover the target clustering using $O(k^2 \log k + k \log n)$ queries, assuming that the clustering satisfies the $\gamma$-margin property for $\gamma > 1$. Now assume that the oracle conforms to the optimal $k$-means clustering solution. In this case, for $1 < \gamma \le \sqrt{3.4} \approx 1.84$, solving $k$-means clustering would be NP-hard without queries, while it becomes efficiently solvable with the help of an oracle [5].

Given a set of instances $\mathcal{X} \subset \mathbf{R}^d$, the $k$-means clustering problem is to find a clustering $\mathcal{C} = \{C_1, \ldots, C_k\}$ which minimizes $f(\mathcal{C}) = \sum_{C_i} \min_{\mu_i \in \mathbf{R}^d} \sum_{x \in C_i} \|x - \mu_i\|_2^2$. The decision version of $k$-means is, given some value $L$, is there a clustering $\mathcal{C}$ with cost $\le L$? The following theorem is the main result of this section.

**Theorem 10.** *Finding the optimal solution to Euclidean $k$-means objective function is NP-hard when $k = \Theta(n^\epsilon)$ for any $\epsilon \in (0, 1)$, even when the optimal solution satisfies the $\gamma$-margin property for $\gamma = \sqrt{3.4}$.*

This results extends the hardness result of [BDR14] to the case of Euclidean metric, rather than arbitrary one, and to the $\gamma$-margin condition (instead of the $\alpha$-center proximity there). The full proof is rather technical and is deferred to the supplementary material (appendix C).

### 4.1.1 Overview of the proof

Our method to prove Thm. 10 is based on the approach employed by [Vat09]. However, the original construction proposed in [Vat09] does not satisfy the $\gamma$-margin property. Therefore, we have to modify the proof by setting up the parameters of the construction more carefully.

To prove the theorem, we will provide a reduction from the problem of Exact Cover by 3-Sets (X3C) which is NP-Complete [GJ02], to the decision version of $k$-means.

**Definition 11** (X3C). *Given a set $U$ containing exactly $3m$ elements and a collection $\mathcal{S} = \{S_1, \ldots, S_l\}$ of subsets of $U$ such that each $S_i$ contains exactly three elements, does there exist $m$ elements in $\mathcal{S}$ such that their union is $U$?*

We will show how to translate each instance of X3C, $(U, \mathcal{S})$, to an instance of $k$-means clustering in the Euclidean plane, $X$. In particular, $X$ has a grid-like structure consisting of $l$ rows (one for each $S_i$) and roughly $6m$ columns (corresponding to $U$) which are embedded in the Euclidean plane. The special geometry of the embedding makes sure that any low-cost $k$-means clustering of the points (where $k$ is roughly $6ml$) exhibits a certain structure. In particular, any low-cost $k$-means clustering could cluster each row in only two ways; One of these corresponds to $S_i$ being included in the cover, while the other means it should be excluded. We will then show that $U$ has a cover of size $m$ if and only if $X$ has a clustering of cost less than a specific value $L$. Furthermore, our choice of embedding makes sure that the optimal clustering satisfies the $\gamma$-margin property for $\gamma = \sqrt{3.4} \approx 1.84$.

### 4.1.2 Reduction design

Given an instance of X3C, that is the elements $U = \{1, \ldots, 3m\}$ and the collection $\mathcal{S}$, we construct a set of points $X$ in the Euclidean plane which we want to cluster. Particularly, $X$ consists of a set of points $H_{l,m}$ in a grid-like manner, and the sets $Z_i$ corresponding to $S_i$. In other words, $X = H_{l,m} \cup (\cup_{i=1}^{l-1} Z_i)$.

The set $H_{l,m}$ is as described in Fig. 1. The row $R_i$ is composed of $6m + 3$ points $\{s_i, r_{i,1}, \ldots, r_{i,6m+1}, f_i\}$. Row $G_i$ is composed of $3m$ points $\{g_{i,1}, \ldots, g_{i,3m}\}$. The distances between the points are also shown in Fig. 1. Also, all these points have weight $w$, simply meaning that each point is actually a set of $w$ points on the same location.

Each set $Z_i$ is constructed based on $S_i$. In particular, $Z_i = \cup_{j \in [3m]} B_{i,j}$, where $B_{i,j}$ is a subset of $\{x_{i,j}, x'_{i,j}, y_{i,j}, y'_{i,j}\}$ and is constructed as follows: $x_{i,j} \in B_{i,j}$ iff $j \notin S_i$, and $x'_{i,j} \in B_{i,j}$ iff $j \in S_i$. Similarly, $y_{i,j} \in B_{i,j}$ iff $j \notin S_{i+1}$, and $y'_{i,j} \in B_{i,j}$ iff $j \in S_{i+1}$. Furthermore, $x_{i,j}, x'_{i,j}, y_{i,j}$ and $y'_{i,j}$ are specific locations as depicted in Fig. 2. In other words, exactly one of the locations $x_{i,j}$ and $x'_{i,j}$, and one of $y_{i,j}$ and $y'_{i,j}$ will be occupied. We set the following parameters.

$$h = \sqrt{5}, d = \sqrt{6}, \epsilon = \frac{1}{w^2}, \lambda = \frac{2}{\sqrt{3}}h, k = (l-1)3m + l(3m+2)$$

$$L_1 = (6m+3)wl, L_2 = 3m(l-1)w, L = L_1 + L_2 - m\alpha, \alpha = \frac{d}{w} - \frac{1}{2w^3}$$

**Lemma 12.** *The set $X = H_{l,n} \cup Z$ has a $k$-clustering of cost less or equal to $L$ if and only if there is an exact cover for the X3C instance.*

**Lemma 13.** *Any $k$-clustering of $X = H_{l,n} \cup Z$ with cost $\leq L$ has the $\gamma$-margin property where $\gamma = \sqrt{3.4}$. Furthermore, $k = \Theta(n^\epsilon)$.*

The proofs are provided in Appendix C. Lemmas 12 and 13 together show that $X$ has a $k$-clustering of cost $\leq L$ satisfying the $\gamma$-margin property (for $\gamma = \sqrt{3.4}$) if and only if there is an exact cover by 3-sets for the X3C instance. This completes the proof of our main result (Thm. 10).

### 4.2 Lower Bound on the Number of Queries

In the previous section we showed that $k$-means clustering is NP-hard even under $\gamma$-margin assumption (for $\gamma < \sqrt{3.4} \approx 1.84$). On the other hand, in Section 3 we showed that this is not the case if the algorithm has access to an oracle. In this section, we show a lower bound on the number of queries needed to provide a polynomial-time algorithm for $k$-means clustering under margin assumption.

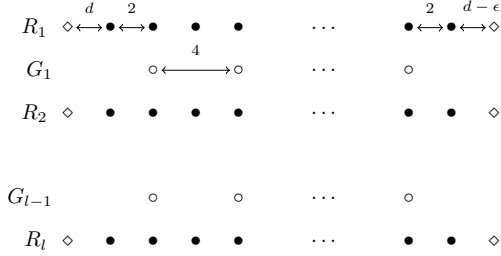

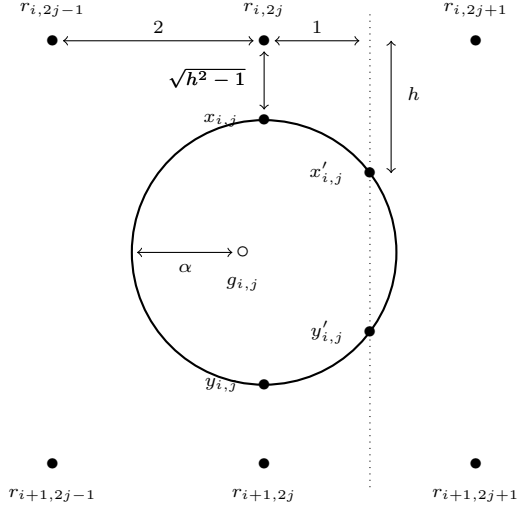

Figure 1: Geometry of $H_{l,m}$. This figure is similar to Fig. 1 in [Vat09]. Reading from letf to right, each row $R_i$ consists of a diamond ($s_i$), $6m + 1$ bullets ($r_{i,1}, \ldots, r_{i,6m+1}$), and another diamond ($f_i$). Each rows $G_i$ consists of $3m$ circles ($g_{i,1}, \ldots, g_{i,3m}$).

Figure 2: The locations of $x_{i,j}$, $x'_{i,j}$, $y_{i,j}$ and $y'_{i,j}$ in the set $Z_i$. Note that the point $g_{i,j}$ is not vertically aligned with $x_{i,j}$ or $r_{i,2j}$. This figure is adapted from [Vat09].

**Theorem 14.** *For any $\gamma \leq \sqrt{3.4}$, finding the optimal solution to the k-means objective function is NP-Hard even when the optimal clustering satisfies the $\gamma$-margin property and the algorithm can ask $O(\log k + \log |\mathcal{X}|)$ same-cluster queries.*

*Proof.* Proof by contradiction: assume that there is polynomial-time algorithm $\mathcal{A}$ that makes $O(\log k + \log |\mathcal{X}|)$ same-cluster queries to the oracle. Then, we show there exists another algorithm $\mathcal{A}'$ for the same problem that is still polynomial but uses no queries. However, this will be a contradiction to Theorem 10, which will prove the result.

In order to prove that such $\mathcal{A}'$ exists, we use a 'simulation' technique. Note that $\mathcal{A}$ makes only $q < \beta(\log k + \log |\mathcal{X}|)$ binary queries, where $\beta$ is a constant. The oracle therefore can respond to these queries in maximum $2^q < k^\beta |\mathcal{X}|^\beta$ different ways. Now the algorithm $\mathcal{A}'$ can try to simulate all of $k^\beta |\mathcal{X}|^\beta$ possible responses by the oracle and output the solution with minimum $k$-means clustering cost. Therefore, $\mathcal{A}'$ runs in polynomial-time and is equivalent to $\mathcal{A}$. $\qquad \square$

## 5 Conclusions and Future Directions

In this work we introduced a framework for semi-supervised active clustering (SSAC) with same-cluster queries. Those queries can be viewed as a natural way for a clustering mechanism to gain domain knowledge, without which clustering is an under-defined task. The focus of our analysis was the computational and query complexity of such SSAC problems, when the input data set satisfies a clusterability condition – the $\gamma$-margin property.

Our main result shows that access to a limited number of such query answers (logarithmic in the size of the data set and quadratic in the number of clusters) allows efficient successful clustering under conditions (margin parameter between 1 and $\sqrt{3.4} \approx 1.84$) that render the problem NP-hard without the help of such a query mechanism. We also provided a lower bound indicating that at least $\Omega(\log kn)$ queries are needed to make those NP hard problems feasibly solvable.

With practical applications of clustering in mind, a natural extension of our model is to allow the oracle (i.e., the domain expert) to refrain from answering a certain fraction of the queries, or to make a certain number of errors in its answers. It would be interesting to analyze how the performance guarantees of SSAC algorithms behave as a function of such abstentions and error rates. Interestingly, we can modify our algorithm to handle a sub-logarithmic number of abstentions by chekcing all possible orcale answers to them (i.e., similar to the "simulation" trick in the proof of Thm. 14).

**Acknowledgments**

We would like to thank Samira Samadi and Vinayak Pathak for helpful discussions on the topics of this paper.

## Footnotes

[1]The exact value of such a threshold $\gamma$ depends on some finer details of the clustering task; whether $d$ is required to be Euclidean and whether the cluster centers must be members of $X$.

[2]In particular, the hardness result of [BDR14] relies on the ability to construct non-Euclidean distance functions. Later in this paper, we prove hardness for $\gamma \leq \sqrt{3.4}$ for Euclidean instances.

[3]In fact, our results are all independent of the Euclidean dimension and apply to any Hilbert space.

[4]It corresponds to the constant appeared in generalized Hoeffding inequality bound, discussed in Theorem 12 in appendix D in supplementary materials.

[5]To be precise, note that the algorithm used for clustering with queries is probabilistic, while the lower bound that we provide is for deterministic algorithms. However, this implies a lower bound for randomized algorithms as well unless $BPP \ne P$

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
