[Supplementary Material]

# Clustering with Same-Cluster Queries

## A  Relationships Between Query Models

**Proposition 1.** *Any clustering algorithm that uses only q same-cluster queries can be adjusted to use 2q cluster-assignment queries (and no same-cluster queries) with the same order of time complexity.*

*Proof.* We can replace each same-cluster query with two cluster-assignment queries as in $Q(x_1, x_2) = \mathbb{1}\{Q(x_1) = Q(x_2))\}$. □

**Proposition 2.** *Any algorithm that uses only q cluster-assignment queries can be adjusted to use kq same-cluster queries (and no cluster-assignment queries) with at most a factor k increase in computational complexity, where k is the number of clusters.*

*Proof.* If the clustering algorithm has access to an instance from each of $k$ clusters (say $x_i \in X_i$), then it can simply simulate the cluster-assignment query by making $k$ same-cluster queries ($Q(x) = \arg\max_i \mathbb{1}\{Q(x, x_i)\}$). Otherwise, assume that at the time of querying $Q(x)$ it has only instances from $k' < k$ clusters. In this case, the algorithm can do the same with the $k'$ instances and if it does not find the cluster, assign $x$ to a new cluster index. This will work, because in the clustering task the output of the algorithm is a partition of the elements, and therefore the indices of the clusters do not matter. □

## B  Comparison of $\gamma$-Margin and $\alpha$-Center Proximity

In this paper, we introduced the notion of $\gamma$-margin niceness property. We further showed upper and lower bounds on the computational complexity of clustering under this assumption. It is therefore important to compare this notion with other previously-studied clusterability notions.

An important notion of niceness of data for clustering is $\alpha$-center proximity property.

**Definition 3** ($\alpha$-center proximity [ABS12])**.** *Let $(\mathcal{X}, d)$ be a clustering instance in some metric space M, and let k be the number of clusters. We say that a center-based clustering $\mathcal{C}_\mathcal{X} = \{C_1, \ldots, C_k\}$ induced by centers $c_1, \ldots, c_k \in M$ satisfies the $\alpha$-center proximity property (with respect to $\mathcal{X}$ and k) if the following holds*

$$\forall x \in C_i, i \neq j, \alpha d(x, c_i) < d(x, c_j)$$

This property has been considered in the past in various studies [BL12, ABS12]. In this appendix we will show some connections between $\gamma$-margin and $\alpha$-center proximity properties.

It is important to note that throughout this paper we considered clustering in Euclidean spaces. Furthermore, the centers were not restricted to be selected from the data points. However, this is not necessarily the case in other studies.

An overview of the known results under $\alpha$-center proximity is provided in Table 1. The results are provided for the case that the centers are restricted to be selected from the training set, and also the unrestricted case (where the centers can be arbitrary points from the metric space). Note that any upper bound that works for general metric spaces also works for the Euclidean space.

We will show that using the same techniques one can prove upper and lower bounds for $\gamma$-margin property. It is important to note that for $\gamma$-margin property, in some cases the upper and lower bounds

Table 1: Known results for $\alpha$-center proximity

|  | Euclidean | General Metric |
|---|---|---|
| Centers from data | Upper bound : $\sqrt{2}+1$ [BL12] <br> Lower bound : ? | Upper bound : $\sqrt{2}+1$ [BL12] <br> Lower bound : 2 [BDR14] |
| Unrestricted Centers | Upper bound : $2+\sqrt{3}$ [ABS12] <br> Lower bound : ? | Upper bound : $2+\sqrt{3}$ [ABS12] <br> Lower bound : 3 [ABS12] |

Table 2: Results for $\gamma$-margin

|  | Euclidean | General Metric |
|---|---|---|
| Centers from data | Upper bound : 2 (Thm. 4) <br> Lower bound : ? | Upper bound : 2 (Thm. 4) <br> Lower bound : 2 (Thm. 5) |
| Unrestricted Centers | Upper bound : 3 (Thm. 6) <br> Lower bound : 1.84 (Thm. 10) | Upper bound : 3 (Thm. 6) <br> Lower bound : 3 (Thm. 7) <br> Awasthi |

match. Hence, there is no hope to further improve those bounds unless P=NP. A summary of our results is provided in 2.

## B.1 Centers from data

**Theorem 4.** *Let $(X, d)$ be a clustering instance and $\gamma \geq 2$. Then, Algorithm 1 in [BL12] outputs a tree $\mathcal{T}$ with the following property:*

*Any $k$-clustering $\mathcal{C}^* = \{C_1^*, \ldots, C_k^*\}$ which satisfies the $\gamma$-margin property and its cluster centers $\mu_1, \ldots, \mu_k$ are in $X$, is a pruning of the tree $T$. In other words, for every $1 \leq i \leq k$, there exists a node $N_i$ in the tree $T$ such that $C_i^* = N_i$.*

*Proof.* Let $p, p' \in C_i^*$ and $q \in C_j^*$. [BL12] prove the correctness of their algorithm for $\alpha > \sqrt{2}+1$. Their proof relies only on the following three properties which are implied when $\alpha > \sqrt{2}+1$. We will show that these properties are implied by $\gamma > 2$ instances as well.

- $d(p, \mu_i) < d(p, q)$
  $\gamma d(p, \mu_i) < d(q, \mu_i) < d(p, q) + d(p, \mu_i) \implies d(p, \mu_i) < \frac{1}{\gamma-1} d(p, q)$.
- $d(p, \mu_i) < d(q, \mu_i)$
  This is trivially true since $\gamma > 2$.
- $d(p, \mu_i) < d(p', q)$
  Let $r = \max_{x \in C_i^*} d(x, \mu_i)$. Observe that $d(p, \mu_i) < r$. Also, $d(p', q) > d(q, \mu_i) - d(p', \mu_i) > \gamma r - r = (\gamma - 1)r$.

$\square$

**Theorem 5.** *Let $(\mathcal{X}, d)$ be a clustering instance and $k$ be the number of clusters. For $\gamma < 2$, finding a $k$-clustering of $X$ which satisfies the $\gamma$-margin property and where the corresponding centers $\mu_1, \ldots, \mu_k$ belong to $\mathcal{X}$ is NP-Hard.*

*Proof.* For $\alpha < 2$, [BDR14] proved that in general metric spaces, finding a clustering which satisfies the $\alpha$-center proximity and where the centers $\mu_1, \ldots, \mu_k \in \mathcal{X}$ is NP-Hard. Note that the reduced instance in their proof, also satisfies $\gamma$-margin for $\gamma < 2$. $\square$

## B.2 Centers from metric space

**Theorem 6.** *Let $(X, d)$ be a clustering instance and $\gamma \geq 3$. Then, the standard single-linkage algorithm outputs a tree $\mathcal{T}$ with the following property:*

*Any $k$-clustering $\mathcal{C}^* = \{C_1^*, \ldots, C_k^*\}$ which satisfies the $\gamma$-margin property is a pruning of $T$. In other words, for every $1 \leq i \leq k$, there exists a node $N_i$ in the tree $T$ such that $C_i^* = N_i$.*

*Proof.* [BBV08] showed that if a clustering $C^*$ has the strong stability property, then single-linkage outputs a tree with the required property. It is simple to see that if $\gamma > 3$ then instances have strong-stability and the claim follows. ☐

**Theorem 7.** *Let* $(\mathcal{X}, d)$ *be a clustering instance and* $\gamma < 3$. *Then, finding a* $k$-*clustering of* $X$ *which satisfies the* $\gamma$-*margin is NP-Hard.*

*Proof.* [ABS12] proved the above claim but for $\alpha < 3$ instances. Note however that the construction in their proof satisfies $\gamma$-margin for $\gamma < 3$. ☐

# C  Proofs of Lemmas 12 and 13

In Section 4 we proved Theorem 10 based on two technical results (i.e., lemma 12 and 13). In this appendix we provide the proofs for these lemmas. In order to start, we first need to establish some properties about the Euclidean embedding of $X$ proposed in Section 4.

**Definition 8** ($A$- and $B$-Clustering of $R_i$)**.** *An* $A$-*Clustering of row* $R_i$ *is a clustering in the form of* $\{\{s_i\}, \{r_{i,1}, r_{i,2}\}, \{r_{i,3}, r_{i,4}\}, \dots, \{r_{i,6m-1}, r_{i,6m}\}, \{r_{i,6m+1}, f_i\}\}$. *A* $B$-*Clustering of row* $R_i$ *is a clustering in the form of* $\{\{s_i, r_{i,1}\}, \{r_{i,2}, r_{i,3}\}, \{r_{i,4}, r_{i,5}\}, \dots, \{r_{i,6m}, r_{i,6m+1}\}, \{f_i\}\}$.

**Definition 9** (Good point for a cluster)**.** *A cluster* $C$ *is good for a point* $z \notin C$ *if adding* $z$ *to* $C$ *increases cost by exactly* $\frac{2w}{3}h^2$

Given the above definition, the following simple observations can be made.

- The clusters $\{r_{i,2j-1}, r_{i,2j}\}$, $\{r_{i,2j}, r_{i,2j+1}\}$ and $\{g_{i,j}\}$ are good for $x_{i,j}$ and $y_{i-1,j}$.
- The clusters $\{r_{i,2j}, r_{i,2j+1}\}$ and $\{g_{i,j}\}$ are good for $x'_{i,j}$ and $y'_{i-1,j}$.

**Definition 10** (Nice Clustering)**.** *A* $k$-*clusteirng is nice if every* $g_{i,j}$ *is a singleton cluster, each* $R_i$ *is grouped in the form of either an* $A$-*clustering or a* $B$-*clustering, and each point in* $Z_i$ *is added to a cluster which is good for it.*

It is straightforward to see that a row grouped in a $A$-clustering costs $(6m+3)w - \alpha$ while a row in $B$-clustering costs $(6m+3)w$. Hence, a nice clustering of $H_{l,m} \cup Z$ costs at most $L_1 + L_2$. More specifically, if $t$ rows are group in a $A$-clustering, the nice-clustering costs $L_1 + L_2 - t\alpha$. Also, observe that any nice clustering of $X$ has only the following four different types of clusters.

(1) Type E - $\{r_{i,2j-1}, r_{i,2j+1}\}$
    The cost of this cluster is $2w$ and the contribution of each location to the cost (i.e., $\frac{cost}{\#locations}$) is $\frac{2w}{2} = w$.
(2) Type F - $\{r_{i,2j-1}, r_{i,2j}, x_{i,j}\}$ or $\{r_{i,2j-1}, r_{i,2j}, y_{i-1,j}\}$ or $\{r_{i,2j}, r_{i,2j+1}, x'_{i,j}\}$ or $\{r_{i,2j}, r_{i,2j+1}, y'_{i-1,j}\}$
    The cost of any cluster of this type is $2w(1 + \frac{h^2}{3})$ and the contribution of each location to the cost is at most $\frac{2w}{9}(h^2 + 3)$. This is equal to $\frac{16}{9}w$ because we had set $h = \sqrt{5}$.
(3) Type I - $\{g_{i,j}, x_{i,j}\}$ or $\{g_{i,j}, x'_{i,j}\}$ or $\{g_{i,j}, y_{i,j}\}$ or $\{g_{i,j}, y'_{i,j}\}$
    The cost of any cluster of this type is $\frac{2}{3}wh^2$ and the contribution to the cost of each location is $\frac{w}{3}h^2$. For our choice of $h$, the contribution is $\frac{5}{3}w$.
(4) Type J - $\{s_i, r_{i,1}\}$ or $\{r_{i,6m+1}, f_i\}$
    The cost of this cluster is $3w$ (or $3w - \alpha$) and the contribution of each location to the cost is at most $1.5w$.

Hence, observe that in a nice-clustering, any location contributes at most $\leq \frac{16}{9}w$ to the total clustering cost. This observation will be useful in the proof of the lemma below.

**Lemma 11.** *For large enough* $w = poly(l, m)$, *any non-nice clustering of* $X = H_{l,m} \cup Z$ *costs at least* $L + \frac{w}{3}$.

*Proof.* We will show that any non-nice clustering $C$ of $X$ costs at least $\frac{w}{3}$ more than any nice clustering. This will prove our result. The following cases are possible.

- $C$ contains a cluster $C_i$ of cardinality $t > 6$ (i.e., contains $t$ weighted points)
  Observe that any $x \in C_i$ has at least $t - 5$ locations at a distance greater than 4 to it, and 4 locations

at a distance at least 2 to it. Hence, the cost of $C_i$ is at least $\frac{w}{2t}(4^2(t-5)+2^24)t = 8w(t-4)$. $C_i$ allows us to use at most $t-2$ singletons. This is because a nice clustering of these $t+(t-2)$ points uses at most $t-1$ clusters and the clustering $C$ uses $1+(t-2)$ clusters for these points. The cost of the nice cluster on these points is $\leq \frac{16w}{9}2(t-1)$. While the non-nice clustering costs at least $8w(t-4)$. For $t \geq 6.4 \implies 8(t-4) > \frac{32}{9}(t-1)$ and the claim follows. Note that in this case the difference in cost is at least $\frac{8w}{3}$.

- Contains a cluster of cardinality $t=6$
  Simple arguments show that amongst all clusters of cardinality 6, the following has the minimum cost. $C_i = \{r_{i,2j-1}, r_{i,2j}, x_{i,j}, y_{i-1,j}, r_{i,2j+1}, r_{2j+2}\}$. The cost of this cluster is $\frac{176w}{6}$. Arguing as before, this allows us to use 4 singletons. Hence, a nice cluster on these 10 points costs at most $\frac{160w}{9}$. The difference of cost is at least $34w$.
- Contains a cluster of cardinality $t=5$
  Simple arguments show that amongst all clusters of cardinality 5, the following has the minimum cost. $C_i = \{r_{i,2j-1}, r_{i,2j}, x_{i,j}, y_{i-1,j}, r_{i,2j+1}\}$. The cost of this cluster is $16w$. Arguing as before, this allows us to use 3 singletons. Hence, a nice cluster on these 8 points costs at most $16w\frac{8}{9}$. The difference of cost is at least $\frac{16w}{9}$.
- Contains a cluster of cardinality $t=4$
  It is easy to see that amongst all clusters of cardinality 4, the following has the minimum cost. $C_i = \{r_{i,2j-1}, r_{i,2j}, x_{i,j}, r_{i,2j+1}\}$. The cost of this cluster is $11w$. Arguing as before, this allows us to use 2 singletons. Hence, a nice cluster on these 6 points costs at most $\frac{32w}{3}$. The difference of cost is at least $\frac{w}{3}$.
- All the clusters have cardinality $\leq 3$
  Observe that amongst all non-nice clusters of cardinality 3, the following has the minimum cost. $C_i = \{r_{i,2j-1}, r_{i,2j}, r_{i,2j+1}\}$. The cost of this cluster is $8w$. Arguing as before, this allows us to use at most 1 more singleton. Hence, a nice cluster on these 4 points costs at most $\frac{64w}{9}$. The difference of cost is at least $\frac{8w}{9}$.
  It is also simple to see that any non-nice clustering of size 2 causes an increase in cost of at least $w$.

$\square$

*Proof of lemma 12.* The proof is identical to the proof of Lemma 11 in [Vat09]. Note that the parameters that we use are different with those utilized by [Vat09]; however, this is not an issue, because we can invoke our lemma 11 instead of the analogous result in Vattani (i.e., lemma 10 in Vattani's paper). The sketch of the proof is that based on lemma 11, only nice clusterings of $X$ cost $\leq L$. On the other hand, a nice clustering corresponds to an exact 3-set cover. Therefore, if there exists a clustering of $X$ of cost $\leq L$, then there is an exact 3-set cover. The other way is simpler to proof; assume that there exists an exact 3-set cover. Then, the corresponding construction of $X$ makes sure that it will be clustered *nicely*, and therefore will cost $\leq L$.

$\square$

*Proof of lemma 13.* As argued before, any nice clustering has four different types of clusters. We will calculate the minimum ratio $a_i = \frac{d(y,\mu)}{d(x,\mu)}$ for each of these clusters $C_i$ (where $x \in C_i$, $y \notin C_i$ and $\mu$ is mean of all the points in $C_i$.) Then, the minimum $a_i$ will give the desired $\gamma$.

(1) For Type E clusters $a_i = h/1 = \sqrt{5}$.
(2) For Type F clusters. $a_i = \frac{\frac{\sqrt{4+16(h^2-1)}}{3}}{2h/3} = \sqrt{\frac{17}{5}} \approx 1.84$.
(3) For Type I clusters, standard calculation show that $a_i > 2$.
(4) For Type J clusters $a_i = \frac{2+\frac{\sqrt{6}}{2}}{\frac{\sqrt{6}}{2}} > 2$.

Furthurmore, $|\mathcal{X}| = (12lm + 3l - 6m)w$ and $k = 6lm + 2l - 3m$. Hence for $w = \text{poly}(l, m)$ our hardness result holds for $k = |\mathcal{X}|^\epsilon$ for any $0 < \epsilon < 1$. $\square$

Lemmas 12 and 13 complete the proof of the main result (Thm. 10).

## D Concentration inequalities

**Theorem 12** (Generalized Hoeffding's Inequality (e.g., [AG15])). *Let $X_1, \ldots . X_n$ be i.i.d random vectors in some Hilbert space such that for all $i$, $\|X_i\|_2 \leq R$ and $E[X_i] = \mu$. If $n > c\frac{\log(1/\delta)}{\epsilon^2}$, then with probability atleast $1 - \delta$, we have that*

$$\left\| \mu - \frac{1}{n} \sum X_i \right\|_2^2 \leq R^2 \epsilon$$