[Reviews · NeurIPS 2016]

Reviewer 1

Summary

This paper studies an interactive semi-supervised clustering setting where the learning algorithm is trying to minimize the k-means cost given access to an oracle that can answer pairwise same-cluster queries (i.e. given two points it reveals whether they are in the same cluster in the optimal k-means solution). The paper proves three results. (1) under a margin assumption a randomized polynomial-time algorithm can find the optimum using k^2log(n) queries. (2) even with the margin assumption, without queries minimizing k-means cost is NP hard. (3) Minimizing k-means cost is NP hard even with weaker margin condition and with few queries O(\log(k) + \log(n)).

Qualitative Assessment

I'm curious about the tradeoff between the three knobs in the problem, the margin assumption, the running time, and the number of queries. I wonder if something more general can be proved about this. For example, on one hand, no matter the margin, we can always use O(n^2) queries to solve the problem in polynomial time, while on the other hand, Thm 10 shows that even if the margin is moderately large, if we use no queries the problem is NP-hard. I'm not saying this is a shortcoming of the paper at all, but I wonder whether one could prove something like: for margin gamma, the problem is NP-hard if the number of queries is < f(n,k,gamma) and it is solvable otherwise. I am also curious about what happens if the oracle either makes mistakes (seems natural in many applications) or is misaligned with the k-means optimum. For the former, suppose that the oracle, for each pair of points and with some probability q makes a persistent mistake about whether the pair is in the same cluster or not (by persistent I mean that you cannot resample to denoise the query).For small q, can we design a clustering algorithm in this case? For the latter, suppose the oracle has in mind a clustering with k-means cost that is slightly worse than the optimal. Can we use this oracle to find the _optimal_ k-means solution efficiently? This also seems like it could be possible under the margin assumption, because if the oracle is close to the optimal cost, its clustering must be well aligned with the optimal clustering. It would be cool to explore these questions.

Confidence in this Review

2-Confident (read it all; understood it all reasonably well)


Reviewer 2

Summary

A nice contribution to the semi-supervised clustering literature. A compelling model, especially from the learning theory perspective.

Qualitative Assessment

This paper proposes and analyzes and new clustering model. Basically, in addition to receiving a clustering instance and a distance function, the algorithm can ask for "same cluster" queries, which ask an oracle whether two data-points belong to the same cluster or not. This is a novel model because previously, algorithms have had access to the distance function or to a query oracle, but not typically both. This allows for some interesting results in a "semi-supervised" model of clustering. The main result is the introduction of a margin property, which has a parameter, which makes the clustering instance easier (in other words the assumption stronger). For k-means, there is a sharp threshold in the value of this parameter. The algorithm is fairly straightforward -- with a sampling and a binary search phase. From my understanding, because of the relationship between this notion and center proximity and careful transfer of lower bounds, this paper proves that there are regimes in which the problem is NP hard without queries, but easy with this additional assumption . My one worry about this model is that the instances where the property is likely to be most helpful (over the unsupervised case) may precisely be the exact instances where it's hardest for the user to tell what clustering they want in the end. However, I still like the direction of this model and the learning-theory contribution of this paper.

Confidence in this Review

2-Confident (read it all; understood it all reasonably well)


Reviewer 3

Summary

The paper looks at center-based clustering in Euclidean space. While it is known that the general k-means (or k-median) problem is hard in this setting, it is also known that a variety of "instance niceness" lead to poly-time clustering algorithms. The paper adds to this literature by examining the case where the instance is also "nice" *and* the algorithm has oracle access to a domain expert that is able to say whether two datapoints reside in the same cluster or not. The authors introduce an alternative notion of niceness, which they call "\gamma-margin" (Def. 1) which basically states for the every cluster center c_i, any point x in cluster i and any point y not-in cluster i we have d(y,c_i)>\gamma d(x,c_i). Then, they provide the following two results for such nice inputs: 1. (upper bound) give an efficient algorithm making O(k^2log(k) + k log(n)) oracle queries and clusters the instance perfectly. 2. (lower bound) prove that no poly-time algorithm with no access to an oracle can cluster a \sqrt{3.4}-margin instance. As a result, no poly-time algorithm can cluster a \sqrt{3.4}-margin instance with log(n) oracle calls (as one can naively traverse all possible the answers to these queries in poly-time).

Qualitative Assessment

This is a great paper that takes a refreshing and a novel view on an interesting line of works, and while the results are not necessarily hard technically, they are still of major significance to the field. I therefore recommend acceptance and even further - a presentation, as it can stir up an interesting discussion and lead to many follow up works. I know that I for one had many interesting follow-up questions: can such an oracle be used with other stability assumptions? what if the domain expert thinks of a clustering which is not center-based but can be well point-wise approximated by a center-based clustering? can the oracle be used to prune the tree used in [KSS04]'s O(n * exp(k)) approximation of the k-means problem? My one major complaint is regarding the lower bound presented in the paper - which is confusing, even when one delves in the cumbersome notation. I believe that there are a few technical errors there and presentation errors that subtract from the quality of the paper and the fact that it is extremely well-written otherwise. I expect the authors' rebuttal to include answers to the following questions, as I am afraid they might lead to substantial changes to the proof of the lower bound: (1) Z_l is ill-defined (you cannot define Z_l using S_{l+1} (lines 261-262) (2) Figure 1 uses "d-\epsilon" rather than "d-\alpha" (unless I didn't understand the role of \alpha) (3) It is my understanding the L_1 is the cost of a type B cluster *per R_i* row (line 131 of appendix) and L_1-\alpha per *R_i* row with type A clustering. And so, I believe L should include a term of L_1*l. (4) As someone who is not familiar with [Vat09], I do not see the "point" of the construction. Namely -- what should "yes" instances be mapped to? Is it to m type-A rows (indicating which sets are selected) and (l-m)-type B rows (the l-m sets not selected)? Or is it a certain selection of the B_{i,j} sets that captures which sets are taken and which aren't? (5) The construction itself is somewhat unclear. In particular, what is w? Is it \eps^{-0.5} (with \eps taken as a parameter) or some large poly(m,l) as stated in the supplementary material? I am unclear as to what is the value of k = O(n^\eps) in this construction... In other words, try to rewrite the construction s.t. it would be clear what the algorithm takes as input, and what is its output. (6) I'd appreciate it if you specifically mention the caveats in the lower bound (actually, this goes to Appendix B as well, as some constructions use Steiner points etc.). In particular, this construction places multiple points at the same location, resulting in an instance with infinite (or exponentially high if you allow miniature points perturbations), but it leaves the question of hard instances with bounded aspect ratio open... Small comments (not to mention - nitpicking comments): * Lines 152-153 should be mentioned explicitly in the intro, or even written in an "Assumption" bulletin. * Algorithm 1 also takes as parameter \beta (or sets \beta as some sufficiently large constant, say 20.) In which case the Algorithm 1's input should be "a parameter" and not "parameters". * Can the (\gamma-1)^{-4} terms be improved to (\gamma-1)^{-2} if you use Chernoff bounds rather than Hoeffding? (i.e. multiplicative bounds on the expected value?) * Line 213 "of the oracle" => "of an oracle" * Line 217 "Section 3" (capital letter) * Line 220 "without query" => "without queries" * Suggestions: use the canonical \textsc{} to denote the X3C problem. * Section 4.1.2 - use \mathcal{X} like used at the rest of the paper. * Thm 14: as k<|X| we have that this is \log(n) effectively. Also, I am not a fan of using big-O notation in a lower bound theorem (but the theorem statement is clear).

Confidence in this Review

3-Expert (read the paper in detail, know the area, quite certain of my opinion)


Reviewer 4

Summary

The paper looks at a type of clustering problem in which there is some margin between points that belong in different clusters, and where there is an oracle that can check whether any two points are supposed to be in the same cluster. They give a probabilistic polynomial-time algorithm for this problem using a bounded number of queries to the oracle. They also prove that the k-means objective is NP-hard for instances with gamma margins below ~1.84, and give a lower bound on the number of queries to the oracle are required for such cases to be solvable in poly-time.

Qualitative Assessment

The writing and notation is a bit unclear/needlessly complicated at times. There are few minor grammatical errors, and some sections are a bit hard to read. The manuscript also appeared not to have been spell checked. The algorithm uses an interesting application of binary search, although a bad choice of estimator for the cluster centre could result in either false negatives or false positives. The authors don't really discuss this point, which I think is important enough that it should be acknowledged. The description of the algorithm (on page 4) reads a bit differently from the actual algorithm on page 5, although this is mostly due to confusing writing. The description seems to state that they select points and stop when one of the clusters is big enough. The algorithm description states that it selects a fixed number of points such that one of the clusters is guaranteed to be big enough. Not an important distinction, but it would be nice if the description were clearer. They claim several times that the algorithm is guaranteed to run successfully when the gamma condition is met, but I'm not entirely convinced based on what is written. The proof of lemma 5 assumes that the difference between the real and sample centres is always less than \eps*radius. This is only proven in lemma 6 (which is confusing - these results need to be rearranged) but it's only a probabilistic bound - so I would assume lemma 5 should be probabilistic too. The overall result in Theorem 7 is with probability 1-\delta, which is weaker than the "guaranteed success" that is asserted in both the abstract and intro. Unless this result follows from something I'm missing, in which case the details should be pointed out somewhere a bit more obvious. Section 4.1 is a bit terse, although I haven't looked at the supplementary content it relies upon. A briefer presentation in the "proceedings" version, without excessive notation only referred to in the supplementary material, would would be clearer. The proof of the query lower-bound is neat. Overall the paper is in need of some editing and a few more details explaining their results. I think the main result might be a bit weaker than is claimed (probabilistic rather than guaranteed, unless I'm overlooking something) and lemmas 5 and 6 need to be reordered and expanded on a bit.

Confidence in this Review

2-Confident (read it all; understood it all reasonably well)


Reviewer 5

Summary

The authors propose a framework for semi-supervised active clustering framework whereby the learner is given oracle access to the domain expert via same-cluster queries. If the expert has decided on a k-center based solution with a certain separability, the authors propose an efficient probabilistic polynomial-time algorithm to recover the optimal clustering. The authors show that there are separable instances where the k-means clustering is NP-hard, but can be efficiently solved given access to a relatively small number of same-cluster queries. Interestingly, the query complexity is not too far away from the lower bound provided by the authors.

Qualitative Assessment

This paper is very well written, clearly motivated and provides substantial theoretical insight into a difficult problem. I enjoyed reading it. I particularly liked the phase transition behaviour of the k-means objective with respect to the instance niceness. The fact that a relatively small number of same-cluster queries can be used to efficiently solve an otherwise NP-hard problem is encouraging. It would be great if you could address the following comments: 1. It is assumed that the oracle is perfect: A more realistic oracle would be allowed to make mistakes. It's critical to understand what kind of assumptions on the oracle are necessary for the problem to remain efficiently solvable. 2. How much does one gain in practice? Assume that the data is generated by some probabilistic model and satisfies the required separation condition. It would be interesting to compare the time-quality tradeoff obtained by some k-Means approximation (say k-means++) and the proposed algorithm. In practice, it may be the case that k-means++ works extremely well given the same separability assumption. 3. It is assumed that the expert conforms to a hard center-based clustering. Do any of the results translate to the setting in which the goal is to recover a GMM and the oracle returns a probabilistic assignment to each cluster? Clearly, given enough separation and some assumptions on the covariance matrices, the problems become very similar. Minor comment: Theorem 7, "clustering instane" should be "clustering instance".

Confidence in this Review

2-Confident (read it all; understood it all reasonably well)


Reviewer 6

Summary

The authors detail a framework for clustering with same-clustering queries and show

Qualitative Assessment

Larger issues: 1. The paper is missing an important piece: empirical studies to demonstrate (at least some of) the various theoretical properties of the SSAC algorithm on real datasets. This is a major issue and should be included in any revision of the paper, even if it's just in the supplementary material. 2. The authors state that they focus on "center-based clustering in Euclidean space", but that the results apply to any Hilbert space. A further discussion (or proof?) of this should be included. Similarly, the authors only discuss K-means. Is the use of this approach limited to K-means? The authors should discuss potential usage with other types of clustering techniques. 3. The theoretical results in Section 3 seem awkwardly placed, without any introductory statements or discussion of these properties in context. The authors should add some transitional / surrounding text here, and (ideally) split the theoretical properties into their own subsection. 4. Some of the proofs aren't really proofs. E.g., for Lemma 6, Theorem 14, and some others, the proofs seem a bit scattered and leave out many important details. Be sure to provide all fully-detailed proofs in the supplementary materials Other issues: -- There are a lot of typos. Here is a non-exhaustive list that I found: "address this issues"; "exhibit some nice properties and likely to"; "In such cases, optimal"; "help relaxing"; remove comma in "framework and, the rather unusual"; additional unnecessary parentheses around [ABS12]; "interactiveness" should be "interactivity"; "niceness of data under"; "Voroni" should be "Voronoi"; "section", "appendix", etc should be capitalized; subsection titles should have consistent capitalization (refer to NIPS guidelines here); "be a clustering instane"; "This results extends"; "which" is overly misused and should be "that" in several places; and possibly some others. -- The term "niceness" should be defined early in the document; the authors should be careful not to confuse their usage of this with the colloquial usage of nice, as in the authors own phrase, "some nice properties". -- Does "success" in the abstract refer to the completion of the algorithm or the accuracy of the algorithm? If accuracy, how are you defining success? -- Section 1.1 has some repeated information from earlier in the paper -- Why is 3.4 such a special number? The authors should explain why this is used in more detail. -- The authors should provide some additional intuition on the equivalency of Definition 4 and their definition of a same-cluster query in Definition 2. This is mostly clear in the text right now, but a brief example would help enhance the reader's understanding. -- "the solution of choice may vary significantly between different intended applications." Can the authors provide references for this and the subsequent sentence?

Confidence in this Review

2-Confident (read it all; understood it all reasonably well)